# Associations of Grazing and Rumination Behaviours with Performance Parameters in Spring-Calving Dairy Cows in a Pasture-Based Grazing System

**DOI:** 10.3390/ani13243831

**Published:** 2023-12-12

**Authors:** Muhammad Wasim Iqbal, Ina Draganova, Patrick Charles Henry Morel, Stephen Todd Morris

**Affiliations:** School of Agriculture and Environment, College of Sciences, Massey University, Private Bag 11-222, Palmerston North 4442, New Zealand; i.draganova@massey.ac.nz (I.D.); p.c.morel@massey.ac.nz (P.C.H.M.); s.t.morris@massey.ac.nz (S.T.M.)

**Keywords:** grazing time, rumination time, individual animal data, grazing dairy cows, production parameters

## Abstract

**Simple Summary:**

This study explored the associations of the lengths of time spent grazing and ruminating with performance parameters (milk yield, milk fat, milk protein, milk solids, liveweight, and the body condition score) in grazing dairy cows. The study selected a group of grazing dairy cows of different breeds and milking ages to record individual animals’ grazing and ruminating behaviours and performance parameters over three consecutive milking seasons. To record behaviour, an automated device AfiCollar was used,. The cows were mainly offered grass, with some additional feeds in various seasons, and milked once a day at 5:00 h. The behavioural variables showed weak to moderate relationships with the performance parameters. Compared with the grazing time, rumination time explained a larger proportion of the variance in milk yield, milk fat, milk protein, and milk solids. These findings support the recording of grazing and rumination behaviours to evaluate the milk yield and its composition.

**Abstract:**

This study investigated the relationship of the length of time spent grazing and ruminating with the performance parameters of spring-calved grazing dairy cows (n = 162) over the lactation period for three lactation seasons (n = 54 per season). The cows were Holstein Friesian (HFR), Jersey (JE), and a crossbreed of Holstein Friesian/Jersey (KiwiCross), with 18 cows from each breed. The cows were either in their 1st, 2nd, 3rd, or 4th lactation year, and had different breeding worth (BW) index values (103 < BW > 151). The cows were managed through a rotational grazing scheme with once-a-day milking in the morning at 05:00 h. The cows were mainly fed on grazed pastures consisting of perennial ryegrass (*Lolium perenne*), red clover (*Trifolium pretense*), and white clover (*Trifolium repens*), and received additional feeds on various days in the summer and autumn seasons. This study used an automated AfiCollar device to continuously record the grazing time and rumination time (min/h) of the individual cows throughout the lactation period (~270 days) for three consecutive years (Year-1, Year-2, and Year-3). The milk yield, milk fat, milk protein, milk solids, liveweight, and body condition score data of the individual animals for the study years were provided by the farm. *PROC CORR* was used in SAS to determine the correlation coefficients (r) between the behaviour and production parameters. A general linear model fitted with breed × lactation year, individual cows, seasons, feed within the season, grazing time, rumination time, as well as their interactions, was assessed to test the differences in milk yield, milk fat, milk protein, milk solids, liveweight, and body condition score. The type I sum of squares values were used to quantify the magnitude of variance explained by each of the study factors and their interactions in the study variables. Grazing time exhibited positive associations with MY (r = 0.34), MF (r = 0.43), MP (r = 0.22), MS (r = 0.39), LW (r = −0.47), and BCS (r = −0.24) throughout the study years. Rumination time was associated with MY (r = 0.64), MF (r = 0.57), MP (r = 0.52), and MS (r = 0.57) in all study years, while there were no effects of rumination time on LW (r = 0.26) and BCS (r = −0.26). Grazing time explained up to 0.32%, 0.49%, 0.17%, 0.31%, 0.2%, and 0.02%, and rumination time explained up to 0.39%, 6.73%, 4.63%, 6.53%, 0.44%, and 0.17% of the variance in MY, MF, MP, MS, LW, and BCS, respectively.

## 1. Introduction

Dairy production in New Zealand is reliant on grazing pasture, which is the cheapest diet and a major source of the nutrient supply for the cows [1,2]. A successful dairy system requires cows capable of consuming a large quantity of quality pasture and converting the ingested feed into high-quality milk [3]. However, the grazing system is characterised by low herbage intake, which is considered a key factor limiting milk production in high-producing cows [4]. Many factors related to the animal itself (e.g., breed and lactation) and its pasture (such as quantity and nutritional value) influence the animal’s ability to meet its nutritional requirements and cause significant variation between individual animals in their nutritional requirements [5,6,7]. In pasture-based systems, dairy cows utilise more than 50% of their daily time grazing and ruminating (up to 15 h per cow per day) [8,9]. Herbage intake is expressed as the product of the grazing time, biting rate, and bite mass, and it is largely regulated by the time spent grazing [10]. Thus, grazing behaviour of animal also has an important role to play in dictating the balance between nutrient intake, energy expenditure, and the energy available for milk production [11]. 

In addition to the factors associated with the environment, grass, and the management of cows [12], milk production in a grazing-based system is limited by the animals’ ability to consume enough herbage [13]. For instance, the behavioural choices made by grazing animals lead to variations in their intake rate [14] and differences in the animals’ milk production. Therefore, grazing behaviour may not only affect the animals’ herbage intake, it can also subsequently influence the animals’ productivity, because the levels of nutrients consumed by the animal effect its milk production [15,16]. One previous study [17] suggested that because of the effect of herbage intake on milk production, the milk yield may be affected by the length of grazing time. Previous studies have reported possible links between grazing behaviour, the milk production level, and the liveweight of dairy cows [18,19]. Likewise, another study found positive relationships between grazing behaviour, milk yield, and the age of the animal [20]. Another study reported that the performance of dairy cows is affected by alterations in their grazing behaviour [21]. These findings further indicate that the influence of grazing behaviour on productivity may be as important as the feed intake.

Milk production and its efficiency are affected not only by the amount and quality of pasture grazed by the cow but also by the way it is digested. In addition to grazing, rumination is a key activity and a trait indicative of appropriate digestive operations in grazing ruminants. Rumination, through regurgitation and remastication, facilitates the accessibility of ingested forage to fermenting bacteria to enhance fibre digestibility [22]. The greater digestibility of dry matter thus enables cows to consume more digestible energy, resulting in greater milk yields [23]. The time a cow spends ruminating has also been found to be positively associated with feed intake [24]. However, cows that spend more time ruminating have less time for grazing and likely to have a lower herbage intake [25]. On the other hand, another study [26] suggested that the feed intake should be considered a measure of the body condition score (BCS) and liveweight (LW) because animals’ feed demands are addressed based on their body weight; this may influence grazing and rumination behaviour, as well as milk production and composition.

In a grazing system, animals adapt their behaviour according to ever-changing sward and weather conditions [27]. The behaviour of cows kept in a grazing herd is usually expressed as the whole herd’s average behaviour and considered an indicator of the external and/or internal environments of animals. Management strategies based on the herd’s average behaviour may not equally benefit all animals, and might have a negative impact on atypical animals [28]. To improve the productivity and welfare of individual animal in a herd, management strategies should focus on individual behavioural trends. Measuring behaviour in individual grazing cows is labour-intensive, and indirect techniques such as visual measurements have several sources of error [10]. Measuring the behaviour of individual dairy cows in a grazing-based system has historically been a challenge because such practices typically involve a large group of animals grazing together. Advanced Precision Livestock Farming (PLF) tools enable the recording of grazing and rumination behaviours of individual animals in a herd.

Understanding the impact of grazing and rumination behaviours on animal productivity could assist in improving the dairy cow management system. The possible relationship between efficiency or productivity and grazing behaviour has attracted considerable interest in recent years [22]. However, such studies have not investigated how individual animals’ behaviour variability relates to variability in their milk production, liveweight (LW), and body condition score (BSC). Likewise, few studies have been published on the associations of BCS and LW with grazing time and rumination time. Thus, the primary objective of this study was to evaluate the possible associations between grazing and rumination behaviours with milk production and composition, liveweight, and body condition score.

## 2. Materials and Methods

### 2.1. Ethical Statement

The study was carried out at Dairy Unit 1, Massey University, Palmerston North, New Zealand (Latitude: −41.3009, Longitude: 174.7720). Approval (Protocol No. 18/58) for the care and handling of animals was received from the Animal Ethics Committee, Massey University, New Zealand.

### 2.2. Grazing Conditions

Dairy Unit 1 operates as a pasture-based farm through a rotational grazing scheme and spring calving system with a once-daily milking practice. The farm spans over 142.7 hectares, divided into 63 paddocks. The local climate follows a temperate pattern with four distinct seasons: spring (September to November), summer (December to February), autumn (March to May), and winter (June to August). Over the study period, the annual average temperature in the area was ~16 °C (8–24 °C) and the annual rainfall was ~960 mm [29].

### 2.3. Animals

This study assessed spring-calved, lactating dairy cows (n = 162) over three years with a random subgroup of 54 cows selected each year (54 × 3 = 162). These selected cows were a subset of the larger herd (n = ~260) at the farm. The selected cows were grazed and managed with the other cows in one herd. Some of the cows were included in the study more than once due to random resampling each year from the herd. The selection criteria considered were breed, lactation year, and breeding worth (BW) of the cows. The breeds were Holstein Friesian (HFR), Jersey (JE), and a crossbreed of Holstein Friesian/Jersey (KiwiCross) with 18 cows per breed (18 × 3 = 54) chosen annually. Within each breed category, 18 cows represented three different lactation years, with 6 cows per lactation year (6 × 3 = 18). The cows were distributed across their 1st, 2nd, 3rd, or 4th year of lactation, and had different breeding worth (103 < BW > 151). The BW of a cow is a metric for its genetic merit for farm profit [30]. The breeding worth is an index that ranks cows on their possible ability to breed profitable, and more efficient, replacements; its calculation involves combining breeding values (cow’s genetic merit for a trait) with their economic values (an estimate of the future dollar value per unit change in a trait) for each trait, and summing them. Throughout the study period, all the cows grazed in the same rotational paddock except for the period when they were brought to the milking shed (~3 h) at 05:00 a.m.

### 2.4. Feeding

The feeding regimen of the study cows over the three years is outlined in Table 1. The main diet of the cows was a pasture of perennial ryegrass (*Lolium perenne* L.) with a mix of red clover (*Trifolium pretense*) and white clover (*Trifolium repens*). Additionally, during the spring season, cows grazed chicory (*Cichorium intybus*). In order to meet their energy requirements and adapt to the seasonal variations in pasture quality and production [31], cows were given additional supplements on some days during the summer and autumn seasons. This included maize (*Zea mays*) silage, corn gluten (*Zea mays* L.), tapioca (*Manihot esculenta*), turnips (*Brassica rapa subsp.* rapa), and distillers’ grains. Supplementary feeds are used when the quality of pasture is low to maintain the cow’s energy intake and production, and also to achieve calving body condition score (BCS) targets [32]. Pellets of distiller’s grain (DG), corn gluten (CG), and ground meal of tapioca were usually fed in the feeding area in the milking shed after milking, whereas maize silage (MS), grass silage (GS), and turnips stem and leaves) were fed at around midday in the paddock. All the cows had equal access to the supplementary feed; it was provided at the same time to all cows. However, the actual intakes of grass or supplements were not recorded or calculated for individual cows. The cows were provided access to drinking water ad libitum in each paddock.

### 2.5. Behaviour Recording

AfiCollar (Afimilk Ltd., Kibbutz, Afikim, Israel), an automated device, continuously monitored and recorded the time spent grazing and ruminating by the individual cows. The collar device has been validated for recording grazing and rumination times in grazing dairy cows [33]. The data recorded by AfiCollar have a strong association and level of agreement with the data obtained through visual observations with a significant (*p* < 0.05) linear relationship for both the grazing time (r = 0.91, CCC = 0.71) and rumination time (r = 0.89, CCC = 0.80).

Throughout the lactation period, the AfiCollar device continuously recorded the minute-by-minute behaviour of grazing cows for 24 h over three consecutive study years. The AfiCollar device has a triaxial (x, y, z) accelerometer-based sensor which is positioned on the right side of the animal’s neck. The sensor has the capability to identify and classify grazing and rumination behaviours depending on the patterns of the animal’s head movements. The built-in generic algorithms analyse the data collected through the sensor and calculate the minutes per hour spent grazing or ruminating by the animal. The recorded data were transmitted wirelessly to the base station, located in the milking shed, through Wi-Fi while cows were in the range of ~500 m. The data were available in a Microsoft Excel spreadsheet (Version 2020, Microsoft Corporation, Redmond, WA, USA. Retrieved from https://office.microsoft.com/excel, accessed on 12 September 2023), and manually downloaded from the computer attached to the base station.

### 2.6. Data Collection and Preparation

#### 2.6.1. Performance Data

Performance data were provided by Dairy Unit 1. Data collection for the milk yield (L), milk fat (kg), milk protein (kg), and milk solids (kg) of the individual cows was performed by the Livestock Improvement Corporation (LIC) through herd testing. Herd tests were conducted by the LIC at Dairy Unit 1 once each month over the whole lactation period. The body condition scores (BCSs) of individual animals were recorded monthly by the farm staff using a 1–5 BCS scale [34]. Liveweight (LW) data of the individual cows were collected monthly by the farm staff using automated Tru-test XR3000 WOW Scales (Tru-Test Pty Ltd., Sunnybank, Australia).

#### 2.6.2. Behaviour Data

The grazing times and rumination times of the individual cows were systematically recorded throughout the whole lactation period for three consecutive years from 2018 to 2021. The lactation period extended from August to April of the following year (~270 days), which is typical for the New Zealand spring calving system. Data collection started for each cow after calving and ended upon drying off. The lactation period encompassed the spring, summer, and autumn seasons. Cows were at the dry stage in winter; thus, no data were collected in this season. The lactation period for 2018–2019 was labelled as Year-1; 2019–2020 was Year-2; and 2020–2021 was Year-3. The AfiCollar originally summarised the behaviour activities as minutes per hour (min/h) spent grazing and ruminating. 

The data collected over the study period were evaluated using summary statistics to identify outliers and any gaps in the automatic data recording. A normality test was carried out to analyse data distribution for the individual animals. The behaviour data for each animal were normally distributed. The farm Standard Operating Procedures for performance data handling were followed during the study.

The daily (min/day) grazing times and rumination times were manually calculated for individual cows using the min/h behaviour data. Subsequently, the monthly averages of grazing and rumination behaviours were calculated for the individual cows for the days when herd testing was performed on the farm. Monthly average grazing times (min/day) and rumination times (min/day) of the individual cows, in addition to the monthly values of each performance variable over the lactation period, were sorted separately for each year over the study period. The data collected were further classified into different breeds, lactation years, seasons, and feed within a season.

### 2.7. Statistical Analysis

The study years differed in supplementary feeds offered and their lactation year; therefore, the significance and effect size of different factors were evaluated separately for the individual year.

#### 2.7.1. Correlation between Behaviour and Performance

Overall Pearson’s correlation coefficients were determined between behaviour (grazing time and rumination time) and performance parameters (milk yield, milk fat, milk protein, milk solids, liveweight, and body condition score) using PROC CORR in SAS (version 9.4, SAS Institute Inc., Cary, NC, USA).

#### 2.7.2. Significance of Study Factors

A general linear model fitted with breed × lactation (with three levels each) and their interactions while accounting for grazing time, rumination time, individual cow, season, feed within the season was performed in SAS (version 9.4, SAS Institute Inc., Cary, NC, USA) using PROC GLM to test the differences in milk yield, milk fat, milk protein, milk solids, liveweight, and body condition score. The milk yield (MY), milk fat (MF), milk protein (MP), milk solids (MS), liveweight (LW), and body condition score (BCS) were used as dependent variables. The main fixed effects were breed, lactation, and their interactions. The individual cows within the breed and lactation year were incorporated as a random effect. Given the length of the lactation period from August to April covering spring, summer, and autumn, the season as a fixed effect and its interactions with other fixed factors were included in the statistical model. Cows received varied supplementary feeds across different seasons within each study year; thus, the feeding regime within the season was also added as a fixed effect. The grazing time (min/day) and rumination time (min/day) were added as continuous covariates. Furthermore, the interactions between the covariates and the fixed effects were also incorporated in the model to assess whether the relationships between dependent variables and covariates varied among the fixed effects. The model used is outlined below:Yijklm = μ + Breedi + Lactationj + Breedi × Lactationj + Cowk (Breei × Lactationj) + Seasonl + Feedm (Seasonl) + Breedi × Seasonl + Lactationj × Seasonl + Breedi × Lactationj × Seasonl + Grazing time (GT)i + Rumination time (RT)j + GTi × Breedi + RTj × Breedi + GTi × Lactationj + RTj × Lactationj + GTi × Breedi × Lactationj + RTj × Breedi × Lactationj + GTi × Seasonl + RTj × Seasonl + ϵijk
where Yijklm represents the ith breed, jth represents lactation, kth represents the cow (breed × lactation), lth represents the season, mth represents the feed (season), μ is the overall mean, and ϵijk is the error term.

#### 2.7.3. The Relative Effect Size of Study Factors

The relative effect sizes and the strength of various study factors and their interactions on the milk yield, milk fat, milk protein, milk solids, liveweight and body condition score were further determined. This was performed using the relative contribution to the total variance (SS Type I) of each study factor and interaction. 

## 3. Results

### 3.1. Correlation between Behaviour and Performance

The correlation coefficient (r) values of grazing time (GT) and rumination time (RT) with milk yield (MY), milk fat (MF), milk protein (MP), milk solids (MS), liveweight (LW), and body condition score (BCS) for different study years (Year-1, Year-2, and Year-3) are shown in Figure 1, Figure 2 and Figure 3. The r values of GT for MY were up to 0.34, MF was up to 0.43, MP was up to 0.22, MS was up to 0.39, LW was down to −0.47, and BCS was down to −0.24. The r values of RT with MY were up to 0.64, with MF up to 0.57, MP was up to 0.52, with MS up to 0.57, LW up to 0.26, and BCS was down to −0.26.

### 3.2. Significance of Study Factors

The breed effect was significant for MY, MP, and MS, but non-significant for MF in all study years (Table 2). The breed effect remained significant for LW (except in Year-2) and non-significant for BCS throughout the study years. The lactation year of the cow affected the MY (except in Year-2), MF (except in Year-2), MP, and MS in all study years. The lactation effect was only significant for LW and BCS in Year-1, while the lactation year had non-significant effects on LW and BCS in Year-2 and Year-3. The individual cow showed significant effects for MY, MF, MP, MS, LW and BCS throughout the study years. The season affected the MY, MF, MP (except in Year-3), and MS in all study years. The season effect was significant for LW in Year-1 and Year-3, and for BCS in all study years. The grazing time only had a significant positive association with the MY, MF, MP, and MS in Year-1. The grazing time had significant negative associations with LW in Year-1 and Year-2, and a significant negative association with BCS in Year-1. The rumination time showed non-significant positive relationships with MY, MF, MP, and MS in all study years. The rumination time only showed a significant positive relationship with LW and a significant negative relationship with BCS in Year-1.

### 3.3. The Relative Effect Size of Study Factors

Grazing time explained 0.11%, 11.3%, and 5.5% of the variance in MY; 2.90%, 18.18%, and 9.46% of the variance in MF; 0.16%, 4.63%, and 3.59% of the variance in MP; and 0.62%, 14.82%, and 7.01% of the variance in MS in Year-1, Year-2, and Year-3, respectively (Table 3). Grazing time explained 9.96%, 4.89%, and 22.14% of the variance in LW and 1.89%, 1.98%, and 5.93% of the variance in BCS in Year-1, Year-2, and Year-3 of the study period, respectively. Rumination time determined 8.76%, 31.20%, and 24.56% of the variance in MY; 5.73%, 19.48%, and 16.68% of the variance in MF; 5.44%, 22.72%, and 21.21% of the variance in MP; and 5.84%, 21.60%, and 20.84% of the variance in MS in Year-1, Year-2, and Year-3, respectively. Rumination time explained 2.96%, 1.85%, and 9.75% of the variance in LW and 0.85%, 0.0%, and 5.51% of the variance in BCS in Year-1, Year-2, and Year-3, respectively. The variance in MY, MF, MP, MS, BW, and BCS explained by interactions of different study factors and covariates was very low (<1%). Only lactation and season interactions explained up to 5% of the variance in milk yield and milk composition in Year-1 and Year-2.

## 4. Discussion

This study evaluated the effects of grazing time, rumination time, breed, lactation year, season, and feed within the season on production parameters (e.g., milk yield, milk fat, milk protein, milk solids, liveweight, and body condition score) in grazing dairy cows. The experimental cows were managed along with other (non-study) cows as a single herd, in a conventional once-a-day milking system. The feeding offered to the animals was different for the individual study years and may explain the variation in the results among different study years. Therefore, this was the reason for analysing the dataset separately for each study year. The season explained a large amount of variance in the performance parameters. Therefore, it can be stated that lactation, in addition to season (lactation year + lactation stage), explained most of the variance in production parameters. Breed was also among the factors explaining most of the variance. These findings are consistent with studies that have previously reported significant differences in the production performance of cows affiliated with different breeds within different parities, and for different stages of lactation [35]. This study also found that milk production and quality varied because of the season; the season also represented different stages of lactation. Several factors contribute to the seasonal variations in milk yield, such as the type, availability, and quality of pasture [36]. The supplementary feeds offered to the experimental animals during different seasons were neither measured for nutritional contents nor their intake. The lack of nutritional content and intake data is a limitation of the study. Such measurements could be conducted in future research because these factors are likely to affect performance as well as behaviour.

The grazing time was positively associated (r = 0.34) with milk yield (MY) and explained up to 0.32% of the variance in MY, although the relationship of GT with MY was only significant in Year-1. This indicates that an increase in grazing time resulted in increased milk production. A previous study [37] has reported a positive relationship between milk yield and grazing time. Another study [38] found that high-producing cows grazed for longer and had greater bite rates and higher intakes than low-producing cows. Herbage intake is the outcome of time spent grazing, bite rate, and intake per bite [39]; therefore, animals grazing longer have the propensity for taking a higher number of bites, thereby consuming a greater amount of pasture (higher dry matter intake), and resulting in higher milk production. A continuous grazing activity period provides a consistent supply of metabolites that have a positive effect on milk synthesis efficiency [40]. In contrast, some reports have found no relationship between grazing behaviour and milk yield [20,41].

Grazing time also showed an effect on milk fat, protein, and solids. A previous study [42] found a significant effect of time at pasture on milk yield, fat, protein, and liveweight. The grazing time had a significant effect (except Year-2) on the milk yield with a positive correlation (r = up to 0.43), and explained up to 0.49% of the variance in the milk fat. Another study [43] also reported positive associations between the time spent eating and milk fat and protein. This is because eating time is linked with dry matter intake, which is highly correlated with the feed conversion efficiency (r = 0.54 to 0.74) in dairy cattle [44]. Continuous feeding with multiple meals per day has already been reported to increase milk fat yield, probably through the stabilisation of ruminal fermentation [45]. Milk fat is the milk component most responsive to nutritional and environmental factors. This may be attributable to the dietary content consumed by the animals. Providing more frequent feeding might have resulted in a lower rumen pH, which likely led to the elevated milk fat yield in this study. 

The effect of grazing time on the milk protein was also significant in Year-1, with an r-value of up to 0.22, explaining 0.49% of the variance in the milk protein. Previous studies have suggested that feeding behaviour is less likely to influence milk protein due to it having no significant association between the two variables [46]. Milk protein concentrations are generally associated with the DMI and energy supply [47]; thus, high protein contents with a longer time at pasture are reasonable [48]. This study found that grazing behaviour had a significant influence on the production of milk solids, aligned with total milk production. Higher milk solids contents have previously been reported in grass-fed cows compared with total mixed-ration-fed cows [49].

Rumination time had a positive correlation with milk yield (r = up to 0.69) and explained up to 0.39% of the variance. A weak correlation (r = 0.30) between both rumination and milk yield phenotypes has already been noted [50]. Another study observed a positive association between rumination time and milk production in grazing dairy cows, and reported improved feed intake and milk production with longer rumination times [51]. A positive relationship between rumination time and dry matter intake has also been reported in dairy cows [52]; a positive relationship between rumination time and milk production may be indirectly related to dry matter intake [53]. Moreover, most of the rumination activity performed by grazing dairy cows generally occurs in a lying posture at night-time. Due to the positive association between rumination and lying times [22], rumination probably impacts milk production [54].

Rumination time explained up to 6.73% of the variance in milk fat. Longer rumination times directly result in better rumen homeostasis and fibre microbial degradation, leading to an increase in fat percentage [55]. The cows had a greater probability of ruminating while producing milk with greater fat contents [56]. Cows spending less time ruminating while lying likely spend less time chewing, resulting in reduced saliva buffering and lower ruminal pH, leading to reduced milk fat production. 

Rumination time was also associated with milk protein. One study has already reported that the milk protein yield was positively associated with the rumination time [57]. In another study, increased milk protein was associated with enhanced chewing activity and greater rumen pH [58]. The increased protein content can be attributed to the energy provided to the udder by the increased propionic acid supply to the rumen from the grass diet [59]. Furthermore, it has also been inferred that pasture-based diets produce milk with more fat and protein [49]. The rumination time showed a positive relationship with the total milk solids (r = up to 0.57). The total average of milk solids contents over lactation has been observed to be significantly higher in cows fed on pastures than in those fed TMR [59]. Increased rumination times might have produced a more favourable ruminal environment for fibre digestion, thereby enhancing the productive response of cows [48,49]. The increase in milk protein output was indeed due to the better use of energy and nitrogen by ruminal microbes, leading to increased microbial protein synthesis [60]. 

Grazing time had a significant negative association with the LW (r = up to −0.47) of animals, accounting for up to 0.2% of the variance in LW over the lactation period. Rumination time only had a positive association with LW (r = up to 0.26) in Year-1, and explained 0.44% of the variance in LW. Both the grazing time (r = up to 0.24) and rumination time (r = up to −0.26) had a significant association with BCS, explaining 0.20% and 0.17% of the variance, respectively. One prior study [61] reported a significant effect of time spent grazing on the live body weight of grazing dairy cows. The DMI is a measure of body weight, controlled by the time spent grazing. This further explains the validity of the association between LW and grazing time. Moreover, [57] observed that when LW and parity were controlled, the DMI was positively associated with feeding time, and tended to be associated with the rumination time and meal frequency. Additionally, LW changes over the length of lactation and reflects differential energy portioning across treatments. This further explains the 0.2% of variation accounted for by grazing time in this study. The association between rumination time and LW was positive in this study: low-ruminating cows had a higher body weight, and vice versa. Similar results have been observed in other studies [53]. In contrast to the present study, a previous study [51] reported that high-ruminating cows were heavier than low-ruminating cows.

## 5. Conclusions

This multifactorial study identified the associations between grazing and ruminating behaviours and milk production, composition, liveweight, and body condition score. Grazing and ruminating behaviours exhibited weak to moderate correlations with milk yield, milk protein, milk fat, milk solids, body weight, and body condition scores in different years of the study. The magnitude of the variance in milk and milk components’ production, liveweight, and BCS explained by behaviour varied between experimental years. Overall, grazing and ruminating behaviours together explained up to 0.66% of the variance in milk yield, 7.22% of the variance in milk fat, 4.67% of the variance in milk protein, 6.73% of the variance in milk solids, 0.44% of the variance in liveweight, and 0.32% of the variance in BCS. Grazing and rumination behaviour together explained a small amount of the total variance in milk yield and its composition. These values show that behaviour has only a limited effect on production parameters.

## Figures and Tables

**Figure 1 animals-13-03831-f001:**
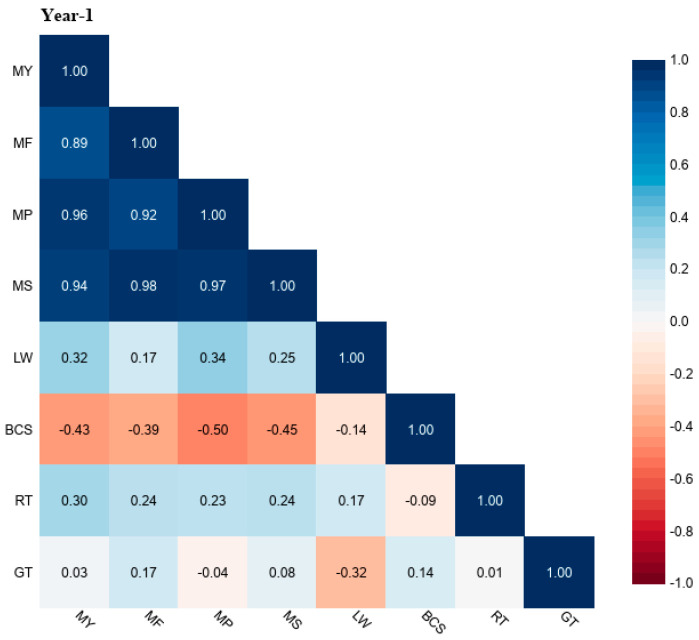
Correlation coefficient (r) values of grazing time (GT) and rumination time (RT) with milk yield (MY), milk fat (MF), milk protein (MP), milk solids (MS), liveweight (LW), and body condition score (BCS) in grazing dairy cows (n = 54) for Year-1 of the study period.

**Figure 2 animals-13-03831-f002:**
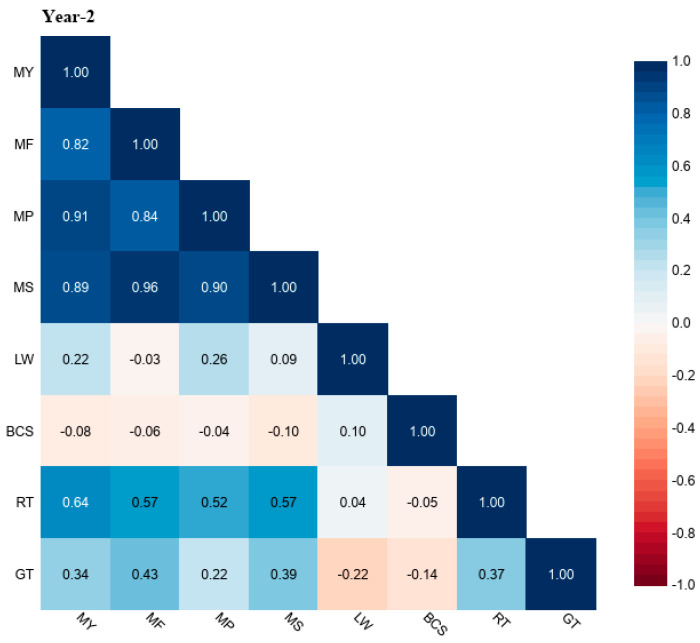
Correlation coefficient (r) values of grazing time (GT) and rumination time (RT) with milk yield (MY), milk fat (MF), milk protein (MP), milk solids (MS), liveweight (LW), and body condition score (BCS) in grazing dairy cows (n = 54) for Year-2 of the study period.

**Figure 3 animals-13-03831-f003:**
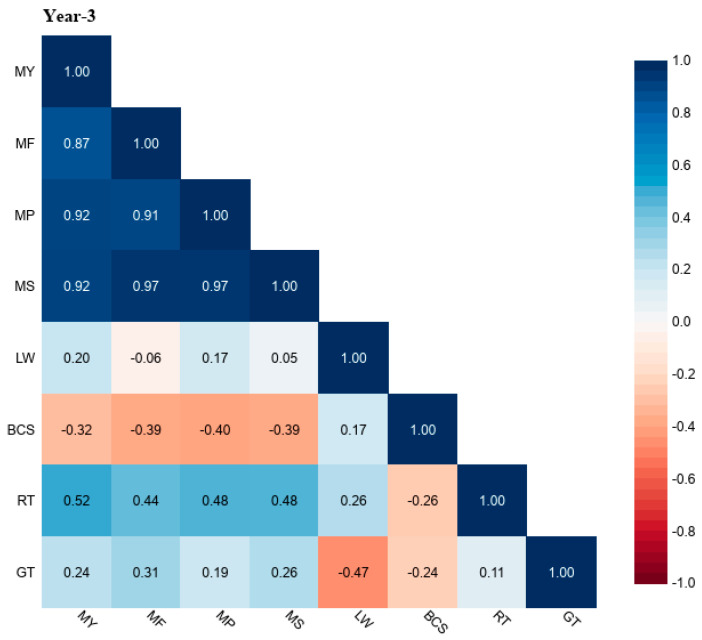
Correlation coefficient (r) values of grazing time (GT) and rumination time (RT) with milk yield (MY), milk fat (MF), milk protein (MP), milk solids (MS), liveweight (LW), and body condition score (BCS) in grazing dairy cows (n = 54) for Year-3 of the study period.

**Table 1 animals-13-03831-t001:** Feeding regimes for grazing dairy cows during different seasons over the study period.

	Feeding Regimes		
Season	Year-1	Year-2	Year-3
Spring	Pasture, Chicory	Pasture, Chicory	Pasture, Chicory
Summer	Pasture, Turnips, Grass Silage	Pasture, Turnips, DDG, Tapioca	Pasture, Turnips, Grass Silage
Autumn	Pasture, Maize Silage	Pasture, Distiller’s Grain, Tapioca	Pasture, Corn Gluten, Tapioca

Year-1, Year-2, and Year-3 are the lactation seasons ranging from 2018 to 2019, 2019 to 2020, and 2020 to 2021, respectively.

**Table 2 animals-13-03831-t002:** Significance (*p* values) for the effects of breed, lactation, cow within breed and lactation, season, supplementary feed within season, grazing time (min/day), rumination time (min/day), and their interactions on milk yield (MY), milk fat (MF), milk protein (MP), milk solids (MS), liveweight (BW), and body condition score (BCS) in Year-1, Year-2, and Year-3 of the study period using a mixed effects model with the individual cow (n = 162) as a random factor, and grazing time and rumination time as covariates.

	Effect	MY	MF	MP	MS	LW	BCS
**Year-1**	Breed	**<0.0001**	0.0928	**0.0004**	0.0099	**<0.0001**	0.1943
	Lactation	**<0.0001**	**<0.0001**	**<0.0001**	**<0.0001**	**0.0004**	**0.0016**
	Breed*Lactation	0.6103	**0.0152**	0.2098	0.0794	0.6808	0.6083
	Cow (Breed*Lactation)	**<0.0001**	**<0.0001**	**<0.0001**	**<0.0001**	**<0.0001**	**<0.0001**
	Season	**<0.0001**	**<0.0001**	**<0.0001**	**<0.0001**	**<0.0001**	**0.0002**
	Feed (Season)	**<0.0001**	**<0.0001**	**<0.0001**	**<0.0001**	**0.0006**	**<0.0001**
	Breed*Season	**<0.0001**	**<0.0001**	**<0.0001**	**<0.0001**	0.0757	0.2719
	Lactation*Season	**<0.0001**	**<0.0001**	**<0.0001**	**<0.0001**	0.1326	0.0137
	Breed*Lactation*Season	**<0.0001**	0.0744	**<0.0001**	**0.0007**	0.1429	0.3501
	Grazing time (GT)	**<0.0001**	**0.0028**	**0.0135**	**0.0023**	**0.0368**	**0.0061**
	Rumination time (RT)	0.7131	0.8873	0.4763	0.8716	**0.0146**	**0.0016**
	GT*Breed	0.3818	0.5803	0.5964	0.5215	0.5604	0.2306
	RT*Breed	0.9574	0.503	0.7612	0.5568	0.1153	**0.0193**
	GT*Lactation	**0.015**	**0.0137**	**0.0144**	**0.0048**	0.6081	0.1963
	RT*Lactation	0.0945	**0.008**	0.0992	**0.0102**	0.7812	**0.0156**
	GT*Breed*Lactation	0.0889	**0.0345**	0.3468	**0.073**	0.9072	0.7579
	RT*Breed*Lactation	0.3328	**0.0333**	0.7599	0.0585	0.4594	0.8792
	GT*Season	0.6434	0.9885	0.7971	0.7299	0.1931	0.2468
	RT*Season	0.9583	0.6177	0.9191	0.7619	0.7258	0.1352
**Year-2**	Breed	**0.0002**	0.2155	**0.0008**	0.057	**<0.0001**	0.1083
	Lactation	0.1353	0.0687	**0.0006**	**0.0077**	**0.0026**	0.2819
	Breed*Lactation	0.5675	0.3669	0.2179	0.4911	0.6377	0.618
	Cow (Breed*Lactation)	0.0884	**<0.0001**	**<0.0001**	**<0.0001**	**<0.0001**	**<0.0001**
	Season	**<0.0001**	**<0.0001**	**<0.0001**	**<0.0001**	**<0.0001**	0.3702
	Feed (Season)	**0.005**	**<0.0001**	**<0.0001**	**<0.0001**	**<0.0001**	**0.0077**
	Breed*Season	0.4436	0.7243	0.0993	0.1288	**<0.0001**	**0.0014**
	Lactation*Season	0.9288	**0.0103**	**<0.0001**	**0.0003**	0.1546	0.9458
	Breed*Lactation*Season	0.9993	0.6054	0.3746	0.1033	0.9701	0.8886
	Grazing time (GT)	0.7005	0.9502	0.2041	0.4987	0.0017	0.0838
	Rumination time (RT)	0.45	0.2634	0.7993	0.6712	0.9428	0.927
	GT*Breed	0.6897	0.0164	0.1518	0.0351	0.6074	0.6865
	RT*Breed	0.0237	0.0964	0.5956	0.1496	0.1649	0.7026
	GT*Lactation	0.639	0.3142	0.9488	0.7112	0.477	0.6385
	RT*Lactation	0.0518	0.2324	**<0.0001**	**0.0073**	0.5152	0.544
	GT*Breed*Lactation	0.4526	**0.0171**	**<0.0001**	**0.0002**	0.4697	**0.0011**
	RT*Breed*Lactation	0.2077	0.2256	0.9089	0.133	0.6104	0.9225
	GT*Season	0.9864	0.0844	0.1267	**0.033**	0.9681	0.9046
	RT*Season	0.7254	0.6409	0.3397	0.5556	0.4077	0.7598
**Year-3**	Breed	**<0.0001**	0.1847	**0.0002**	**0.0074**	**<0.0001**	0.2574
	Lactation	**<0.0001**	**0.0024**	**<0.0001**	**<0.0001**	0.0548	0.0983
	Breed*Lactation	0.1113	0.8728	0.4664	0.7516	0.0804	0.5466
	Cow (Breed*Lactation)	**<0.0001**	**<0.0001**	**<0.0001**	**<0.0001**	**<0.0001**	**<0.0001**
	Feed (Season)	**<0.0001**	**<0.0001**	**<0.0001**	**<0.0001**	**<0.0001**	**<0.0001**
	Season	**<0.0001**	**<0.0001**	**<0.0001**	**<0.0001**	**0.0179**	0.0505
	Breed*Season	**<0.0001**	0.0886	**0.0068**	**0.0157**	**0.0011**	**0.0086**
	Lactation*Season	**0.06**	**0.0054**	**0.0009**	**0.0089**	**0.0114**	0.3075
	Breed*Lactation*Season	0.6791	0.2195	0.881	0.5643	0.3321	0.0533
	Grazing time (GT)	0.0781	0.0457	0.0828	**0.0127**	0.2312	0.2242
	Rumination time (RT)	0.7463	0.8224	0.6707	0.9644	0.6533	0.4508
	GT*Breed	0.4048	0.7453	0.7901	0.9655	0.1441	0.5011
	RT*Breed	0.0826	0.0975	0.1628	0.1296	0.658	0.2601
	GT*Lactation	**0.0008**	0.0684	0.1773	**0.0232**	0.0788	0.3689
	RT*Lactation	**0.0364**	0.1844	**0.019**	**0.0499**	0.8397	0.1508
	GT*Breed*Lactation	0.397	0.5511	0.2489	0.3035	0.461	0.661
	RT*Breed*Lactation	0.6061	0.4369	0.1855	0.4544	0.761	0.8775
	GT*Season	**0.0014**	**0.0065**	**0.0007**	**0.0002**	**0.0329**	**0.0029**
	RT*Season	0.9548	0.4259	0.9157	0.6533	0.5109	0.9616

The significance level was set at a *p*-value of 0.05. * Indicates an interaction between study factors.

**Table 3 animals-13-03831-t003:** The magnitude of variance in milk yield (MY), milk fat (MF), milk protein (MP), milk solids (MS), liveweight (LW), and body condition score (BCS) explained by the effects of breed, lactation year, cow within breed and lactation year, season, and their interactions in three consecutive study years (Year-1, Year-2, and Year-3) using a mixed effects model, with the individual cow (n = 162) as a random factor and the grazing time and rumination time as covariates.

	Effect	MY	MF	MP	MS	LW	BCS
**Year-1**	Breed	11.68	1.78	7.76	3.95	36	4.12
	Lactation	28.21	25.97	37.02	32.97	15.33	15.14
	Breed*Lactation	0.53	6.1	1.91	3.56	2.87	2.51
	Cow (Breed*Lactation)	15.18	18.43	18.41	18.43	36.67	39.79
	Season	27.24	31.93	17.62	26.23	1.49	0.24
	Feed (Season)	2.92	3.52	2.66	2.73	0.37	15.78
	Breed*Season	1.52	0.91	0.98	0.9	0.2	0.47
	Lactation*Season	4.58	2.9	5.05	3.77	0.16	0.85
	Breed*Lactation*Season	0.68	0.28	0.73	0.46	0.22	0.46
	Grazing time (GT)	0.32	0.23	0.17	0.2	0.01	0.15
	Rumination time (RT)	0.34	0.31	0.44	0.42	0.02	0.17
	GT*Breed	0.04	0.05	0.03	0.05	0.04	0.01
	RT*Breed	0.02	0.14	0.03	0.11	0.05	0.31
	GT*Lactation	0.09	0.07	0.07	0.06	0.04	0.13
	RT*Lactation	0.21	0.11	0.12	0.12	0.03	0.61
	GT*Breed*Lactation	0.16	0.17	0.12	0.16	0.05	0.31
	RT*Breed*Lactation	0.14	0.13	0.12	0.15	0.12	0.12
	GT*Season	0.03	0	0.05	0.03	0.12	0.34
	RT*Season	0	0.01	0	0	0.02	0.11
	GT + RT (%)	1.35	1.22	1.15	1.3	0.5	2.26
	Total (%)	94.64	93.59	93.93	94.8	94.01	82.28
**Year-2**	Breed	13.32	2.4	9.44	3.68	38.37	6.39
	Lactation	2.22	3.65	9	6.5	15.42	3.54
	Breed*Lactation	2.11	2.57	3.25	1.98	2.87	3.53
	Cow (Breed*Lactation)	27.37	22.22	19.33	22.17	37.14	46.96
	Season	27.22	40.48	32.98	37.75	1.7	0.01
	Feed (Season)	2.84	12.72	8.33	11.56	1.57	0.92
	Breed*Season	2.04	0.11	0.41	0.26	0.41	2.94
	Lactation*Season	0.31	0.89	1.92	1.07	0.12	0.18
	Breed*Lactation*Season	0.53	0.4	0.44	0.64	0.03	0.45
	Grazing time (GT)	0.02	0.49	0.04	0.2	0	0.2
	Rumination time (RT)	0.39	6.73	4.63	6.53	0.44	0.12
	GT*Breed	0.66	0.63	0.37	0.5	0.01	0.09
	RT*Breed	4.2	0.04	0.22	0.11	0.02	0.11
	GT*Lactation	0.29	0.07	0.03	0.02	0.02	0.03
	RT*Lactation	3.27	0.2	0.99	0.48	0	0.09
	GT*Breed*Lactation	0.76	0.16	0.28	0.34	0.05	1.93
	RT*Breed*Lactation	2.87	0.38	0.08	0.2	0.04	0.1
	GT*Season	0.04	0.2	0.19	0.25	0	0.05
	RT*Season	0.16	0.12	0.08	0.1	0.07	0.1
	GT + RT (%)	12.66	9.02	6.91	8.73	0.65	2.82
	Total (%)	92.18	89.84	92.41	89.66	67.75	98.89
**Year-3**	Breed	9.14	1.28	5.95	3.4	48.39	3.02
	Lactation	9.22	5.06	7.85	7.31	4.41	5.27
	Breed*Lactation	1.57	0.45	1.05	0.59	6.35	3.35
	Cow (Breed*Lactation)	8.84	16.45	13	13.95	31.99	48.54
	Season	43.48	43.54	34.08	39.94	1.32	2.86
	Feed (Season)	8.18	8.86	11.63	10.57	0.19	0.74
	Breed*Season	1.09	0.42	0.78	0.61	0.29	1.07
	Lactation*Season	0.36	0.76	1.03	0.68	0.21	0.37
	Breed*Lactation*Season	0.22	0.55	0.2	0.33	0.14	1.2
	Grazing time (GT)	0.12	0.2	0.16	0.31	0.02	0.12
	Rumination time (RT)	0.28	0.32	0.53	0.5	0.02	0.04
	GT*Breed	0.07	0.03	0.03	0	0.06	0.11
	RT*Breed	0.2	0.24	0.2	0.2	0.01	0.21
	GT*Lactation	0.57	0.28	0.19	0.38	0.08	0.16
	RT*Lactation	0.26	0.17	0.43	0.3	0.01	0.3
	GT*Breed*Lactation	0.16	0.16	0.29	0.24	0.06	0.19
	RT*Breed*Lactation	0.11	0.19	0.34	0.18	0.03	0.09
	GT*Season	0.52	0.52	0.8	0.86	0.11	0.92
	RT*Season	0	0.09	0.01	0.04	0.02	0.01
	GT + RT (%)	2.29	2.2	2.98	3.01	0.42	2.15
	Total (%)	84.72	79.77	78.59	80.42	93.79	69.18

GT + RT represents the variance (%) in milk, milk production, and milk composition explained by the grazing time, rumination time, and their interactions (indicated by a * between two factors) with other study factors. The total represents the total amount of variance (%) explained by all study factors and their interactions. To calculate the variance partitioning for the individual study factors and their interactions, firstly, the total amount (%) of variance explained by the model was calculated using the Type 1 sum of square values. Subsequently, that value was diluted among the individual study factors and their interactions based on their sum of squares of values.

## Data Availability

The data are contained in the manuscript.

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
