# Peer review of "Associations of Grazing and Rumination Behaviours with Performance Parameters in Spring-Calving Dairy Cows in a Pasture-Based Grazing System"

_animals, 2023, doi:10.3390/ani13243831_

Round 1

Reviewer 1 Report

Comments and Suggestions for Authors

This study undertakes an investigation into the intricate relationships between the temporal dynamics of grazing and ruminating behaviors exhibited by grazing dairy cows and their consequential performance parameters, specifically encompassing key metrics such as milk yield, fat content, protein content, milk solids concentration, live weight, and body condition score. This research is underpinned by the inclusion of a diverse cohort of dairy cows representing multiple breeds and spanning various milking ages, observed over the course of three consecutive lactation seasons, facilitated by the deployment of the AfiCollar, an advanced automated behavior monitoring system. The primary contributions of this study are twofold: first, it underscores the discerning revelation that rumination time emerges as a potent explanatory factor accounting for a substantial proportion of the variance observed in performance parameters. Second, it underscores the critical significance of vigilant and continuous monitoring of grazing and rumination behaviors as pivotal determinants in the context of milk yield and its associated compositional constituents. The strengths of this research are grounded in its methodological robustness, manifesting through the acquisition of a rich and comprehensive dataset, the utilization of cutting-edge automated behavioral monitoring technology, and its meticulous consideration of a multitude of factors that collectively shape the performance outcomes of dairy cows.

·      The study doesn't mention the sample size for each of the breed-lactation year combinations. This information is crucial to understand the representativeness of the sample and generalizability of the results.

·      It's unclear how cows were selected for resampling within available cows. The method used for random resampling should be described in more detail.

·      More information about the AfiCollar device's validation for measuring grazing and rumination behaviors should be included. This would help assess the reliability of the data collected using this device.

·      While the manuscript describes the feeding regimes, it does not provide information about the nutritional content of these feeds. The nutritional content is crucial in understanding the impact of different diets on cow performance.

·      The manuscript does not provide information on data cleaning, missing data handling, and quality control measures. It is essential to describe how the data were processed to ensure accuracy.

·      The relevance of the review topic to the field of animal science and dairy farming is clear. Understanding the relationship between grazing, rumination, and dairy cow performance has practical implications for the industry. However, the manuscript should explicitly state the practical implications and potential benefits of the research findings.

·      Authors need to mention adaptation period after mounting AfiCollar device's on experimental animals

Author Response

Comments and Suggestions for Authors

This study undertakes an investigation into the intricate relationships between the temporal dynamics of grazing and ruminating behaviours exhibited by grazing dairy cows and their consequential performance parameters, specifically encompassing key metrics such as milk yield, fat content, protein content, milk solids concentration, live weight, and body condition score. This research is underpinned by the inclusion of a diverse cohort of dairy cows representing multiple breeds and spanning various milking ages, observed over the course of three consecutive lactation seasons, facilitated by the deployment of the AfiCollar, an advanced automated behavior monitoring system. The primary contributions of this study are twofold: first, it underscores the discerning revelation that rumination time emerges as a potent explanatory factor accounting for a substantial proportion of the variance observed in performance parameters. Second, it underscores the critical significance of vigilant and continuous monitoring of grazing and rumination behaviours as pivotal determinants in the context of milk yield and its associated compositional constituents. The strengths of this research are grounded in its methodological robustness, manifesting through the acquisition of a rich and comprehensive dataset, the utilization of cutting-edge automated behavioural monitoring technology, and its meticulous consideration of a multitude of factors that collectively shape the performance outcomes of dairy cows.

RE: Dear reviewer, thank you so much for the time and effort to review our manuscript and provide valuable suggestions. The manuscript has now been revised and the comments have been addressed, details can be found below.

  • The study doesn't mention the sample size for each of the breed-lactation year combinations. This information is crucial to understand the representativeness of the sample and the generalizability of the results.

RE: It has been provided on the lines 151-160 and has been further explained in the Table below.

Year-1

Year-2

Year-3

Breed

Lactation

Lactation

Lactation

Holstein-Friesian × 18

1st × 6

2nd × 6

1st × 6

2nd × 6

3rd × 6

2nd × 6

3rd × 6

4th × 6

3rd × 6

Jersey × 18

1st × 6

2nd × 6

1st × 6

2nd × 6

3rd × 6

2nd × 6

3rd × 6

4th × 6

3rd × 6

HFR-JE Crossbreed × 18

1st × 6

2nd × 6

1st × 6

2nd × 6

3rd × 6

2nd × 6

3rd × 6

4th × 6

3rd × 6

  • It's unclear how cows were selected for resampling within available cows. The method used for random resampling should be described in more detail.

RE: Along with breed and lactation year, a third parameter used to select animals was their breeding worth index value. The criteria are explained below and it remained same for all the study years.

Year-1

Year-2

Year-3

Year-1, 2, 3

Breed

Lactation

Lactation

Lactation

Breeding worth index (BW)

Holstein-Friesian × 18

1st × 6

2nd × 6

1st × 6

Low BW × 2

2nd × 6

3rd × 6

2nd × 6

Medium BW × 2

3rd × 6

4th × 6

3rd × 6

High BW × 2

Jersey × 18

1st × 6

2nd × 6

1st × 6

Low BW × 2

2nd × 6

3rd × 6

2nd × 6

Medium BW × 2

3rd × 6

4th × 6

3rd × 6

High BW × 2

HFR-JE Crossbreed × 18

1st × 6

2nd × 6

1st × 6

Low BW × 2

2nd × 6

3rd × 6

2nd × 6

Medium BW × 2

3rd × 6

4th × 6

3rd × 6

High BW × 2

  • More information about the AfiCollar device's validation for measuring grazing and rumination behaviors should be included. This would help assess the reliability of the data collected using this device.

RE: Validation results have been added, please see the lines 201-206

  • While the manuscript describes the feeding regimes, it does not provide information about the nutritional content of these feeds. The nutritional content is crucial in understanding the impact of different diets on cow performance.

RE: The main objective of this manuscript was to estimate the associations between behaviour and production. Since the animals were fed supplements depending on the season, therefore the Feeding regime within season was added in the model to evaluate the variation explained by the feeding. However, this study did not focus on the effects of feed on performance parameters. Also, we have mentioned lack of nutritional content data as a limitation of the study and something that can be done in future research as it is likely not only to have an effect on performance but also on behaviour. See the lines 437-441.

  • The manuscript does not provide information on data cleaning, missing data handling, and quality control measures. It is essential to describe how the data were processed to ensure accuracy.

RE: The behaviour data were collected continuously using AfiCollar throughout the lactation period. Data evaluation has been mentioned in the manuscript now, see the lines 249-254. On the other hand, the clean performance parameters data were provided by the farm andthe farm has SOP for performance data handling.

  • The relevance of the review topic to the field of animal science and dairy farming is clear. Understanding the relationship between grazing, rumination, and dairy cow performance has practical implications for the industry. However, the manuscript should explicitly state the practical implications and potential benefits of the research findings.

RE: Few sentences have been added in the conclusion, see the lines 523-525

  • Authors need to mention adaptation period after mounting AfiCollar devices on experimental animals

RE: There was no adoption period used in the current study as animals were already adopted. The collars were put on the animals since the calving day and taken off once the animals were dried off.

Reviewer 2 Report

Comments and Suggestions for Authors

The research paper explores the associations of grazing and rumination behaviours with performance parameters in spring-calving dairy cows in a pasture-based grazing system. The performance parameters included milk yield (MY), milk fat (MF), milk protein (MP), milk solids (MS), liveweight (LW) and body condition score (BCS).

The research is worth investigating and the data descriptions are very clear. However, there are certain aspects regarding the models and the results explanations that require further consideration.

Specific comments are outlined below:

1.     The paper mentions that grazing time and rumination time were measured using AfiCollar, validated in a previous paper (lines 179-180). Could the authors provide details on whether the devices were validated for the exact same experiments? For example, was the accuracy of the sensor-derived behaviour classification determined by comparing predicted and observed behaviours? Additionally, it would be valuable to include detailed data on grazing time and rumination time in the paper.

2.     In the Abstract (lines 45-50), correlations between grazing time and MY, MF, MP, MS, LW, and BCS are reported. However, the decision to report the largest absolute values among three years raises questions. Are there specific reasons for this approach? If this approach can be explained then it would be useful for the text to be qualified by stating that the value was the highest of three that were recorded, or something similar.

3.     Furthermore, discrepancies exist between reported and observed correlations in Figures 2 and 3, and the results in Table 3 differ significantly. A careful review of the results is recommended. For example, the correlation between grazing time and BCS is -0.24 rather than 0.24 according to Figure 3. The correlation between rumination time and MY is 0.64 rather than 0.69 according to Figure 2. According to Table 3, grazing time explained up to 0.17% of the variance in MP rather than 0.49%. Similarly, grazing time explained up to 0.02% of the variance in LW rather than 0.2%.

4.     In Lines 251-264, the denotations of the models (e.g., 'Yijklmn') lack clarity. Definitions for variables such as i, j, k, l, m, n and the meaning of 'the ith treatment group A' need clarification.

5.     In line 154-171, the grazing system is described as pasture based but cattle are offered a combination of pasture and supplementary feed. It would be useful to include a description of the relative contribution of pasture and supplements to the diet as this might significantly influence animal behaviour and relationships. For example, a predominantly supplement diet is likely to contain much higher energy per unit of volume than a completely pasture based diet, and this could influence the measure of grazing time required to consume the necessary dietary energetic intake.

6.     Section 2.7.3 (lines 266-271) discusses the relative effect size and variance partitioning. Details on how the results of variance partitioning were obtained should be elaborated. Additionally, addressing the inconsistencies in explanations for Year 1, Year 2, and Year 3 results is crucial.

7.     For Table 2, it is suggested to highlight significant p-values, perhaps using bold or colours for emphasis.

8.     In lines 349-350, the statement "Breed and lactation...explained most of the variation" seems inconsistent with the values in Table 3. According to Table 3, season explained 27.22%, while breed explained 13.32% and lactation explained 2.22% for Year 2. For Year 3, season explained 43.48% while breed explained 9.14% and lactation explained 9.22%. A thorough review and correction of these discrepancies are necessary.

9.     In line 404, it is mentioned that rumination time explained up to 6.73% of the variance in milk fat rather than milk yield. This inconsistency requires clarification.

10. In line 430, clarification is needed for 'DMI,' whether it stands for Dry Matter Intake.

11. Lines 447-449 request a thorough check of the values, exploring how grazing and ruminating behaviours together collectively explain different performance parameters.

12. In line 460, please check the Data Availability Statement.

13. In line 465, please check the Acknowledgments.

Addressing these points will enhance the clarity and accuracy of the research paper.

Author Response

Comments and Suggestions for Authors

The research paper explores the associations of grazing and rumination behaviours with performance parameters in spring-calving dairy cows in a pasture-based grazing system. The performance parameters included milk yield (MY), milk fat (MF), milk protein (MP), milk solids (MS), liveweight (LW) and body condition score (BCS).

The research is worth investigating and the data descriptions are very clear. However, there are certain aspects regarding the models and the results explanations that require further consideration.

RE: Dear reviewer, thank you so much for the time and effort to review our manuscript and provide valuable suggestions. The manuscript has now been revised and the comments have been addressed, details can be found below.

Specific comments are outlined below:

  1. The paper mentions that grazing time and rumination time were measured using AfiCollar, validated in a previous paper (lines 179-180). Could the authors provide details on whether the devices were validated for the exact same experiments? For example, was the accuracy of the sensor-derived behaviour classification determined by comparing predicted and observed behaviours? Additionally, it would be valuable to include detailed data on grazing time and rumination time in the paper.

RE: The data regarding validation has been added, please see the lines 201-206. This experiment was part of the same research project where collar device was validated. The validation experiments were performed earlier year of this study (2019-2020) of the on same animals/or their herd mates grazing under same conditions as described in the current study. The linear relationship was based on the observed and predicted values and reported in the validation paper (https://doi.org/10.3390/ani11092724). Sensitivity specificity, and accuracy were measured and reported in the original draft but later removed as suggested by the reviewers and some some other analysis were included as advised by the reviewers.

  1. In the Abstract (lines 45-50), correlations between grazing time and MY, MF, MP, MS, LW, and BCS are reported. However, the decision to report the largest absolute values among three years raises questions. Are there specific reasons for this approach? If this approach can be explained then it would be useful for the text to be qualified by stating that the value was the highest of three that were recorded, or something similar.

RE: Reporting the strongest values was to provide the limit of maximum effect of behaviour on certain production parameter. This could be helpful in terms of practical implications of the inferences reported in our study.

  1. Furthermore, discrepancies exist between reported and observed correlations in Figures 2 and 3, and the results in Table 3 differ significantly. A careful review of the results is recommended. For example, the correlation between grazing time and BCS is -0.24 rather than 0.24 according to Figure 4, The correlation between rumination time and MY is 0.64 rather than 0.69 according to Figure 2. According to Table 3, grazing time explained up to 0.17% of the variance in MP rather than 0.49%. Similarly, grazing time explained up to 0.02% of the variance in LW rather than 0.2%.

RE: Revised accordingly, see the lines 49-55

  1. In Lines 251-264, the denotations of the models (e.g., 'Yijklmn') lack clarity. Definitions for variables such as i, j, k, l, m, n and the meaning of 'the ith treatment group A' need clarification.

RE: Revised see the lines 316

  1. In line 154-171, the grazing system is described as pasture based but cattle are offered a combination of pasture and supplementary feed. It would be useful to include a description of the relative contribution of pasture and supplements to the diet as this might significantly influence animal behaviour and relationships. For example, a predominantly supplement diet is likely to contain much higher energy per unit of volume than a completely pasture based diet, and this could influence the measure of grazing time required to consume the necessary dietary energetic intake.

RE: The main objective of this manuscript was to estimate the associations between behaviour and production. Since the animals were fed supplements depending on the season, therefore the Feeding regime within season was added in the model to evaluate the variation explained by the feeding. However, this study did not focus on the effects of feed on performance parameters. Therefore, the relative time spent on feeding supplements and grazing was not measured. This has been added a s a limitation of this study as well and has been suggested for future research. Please see the lines 437-441

  1. Section 2.7.3 (lines 266-271) discusses the relative effect size and variance partitioning. Details on how the results of variance partitioning were obtained should be elaborated. Additionally, addressing the inconsistencies in explanations for Year 1, Year 2, and Year 3 results is crucial.

RE: It has been added in the footnotes of Table 3

  1. For Table 2, it is suggested to highlight significant p-values, perhaps using bold or colours for emphasis.

RE: Significant values have been made bold.

  1. In lines 349-350, the statement "Breed and lactation...explained most of the variation" seems inconsistent with the values in Table 3. According to Table 3, season explained 27.22%, while breed explained 13.32% and lactation explained 2.22% for Year 2. For Year 3, season explained 43.48% while breed explained 9.14% and lactation explained 9.22%. A thorough review and correction of these discrepancies are necessary.

RE: corrected, see the lines 428

  1. In line 404, it is mentioned that rumination time explained up to 6.73% of the variance in milk fat rather than milk yield. This inconsistency requires clarification.

RE: corrected, see the lines 479

  1. In line 430, clarification is needed for 'DMI,' whether it stands for Dry Matter Intake.

RE: Revised, see the lines 442

  1. Lines 447-449 request a thorough check of the values, exploring how grazing and ruminating behaviours together collectively explain different performance parameters.

RE: Corrected

  1. In line 460, please check the Data Availability Statement.

RE: Corrected

  1. In line 465, please check the Acknowledgments.

RE: Corrected

Addressing these points will enhance the clarity and accuracy of the research paper.

Reviewer 3 Report

Comments and Suggestions for Authors

Thank you for very interesting article that concerns use of greenlands for milk production and cows behaviour during the grazing and ruminating. Practically i haven't remarks exept one. I would refrain from describing the results in lines 38-44, which are included later in the publication. I would only leave the data that is described in lines 45-50 and is more important for the abstract of the article. The provided research is aimed at new directions in milk production using grasslands and grazing dairy cows. Ruminants should not, in principle, compete with humans for food. Giving up growing corn for silage or feeding grain concentrates would be the best solution to this problem.

Author Response

Comments and Suggestions for Authors

Thank you for very interesting article that concerns use of green lands for milk production and cows’ behaviour during the grazing and ruminating. Practically i haven't remarks except one.

RE: Dear reviewer, thank you so much for the time and effort to review our manuscript and provide valuable suggestions. The manuscript has now been revised and the comments have been addressed, details can be found below.

I would refrain from describing the results in lines 38-44, which are included later in the publication. I would only leave the data that is described in lines 45-50 and is more important for the abstract of the article. 

RE: Removed as advised, see the lines 42-48

The provided research is aimed at new directions in milk production using grasslands and grazing dairy cows. Ruminants should not, in principle, compete with humans for food. Giving up growing corn for silage or feeding grain concentrates would be the best solution to this problem.

Round 2

Reviewer 1 Report

Comments and Suggestions for Authors

The authors addressed all my comments satisfactorily